# Multiple Kernel Clustering with Shifted Laplacian on Grassmann Manifold

## ABSTRACT

Multiple kernel clustering (MKC) has garnered considerable attention, as their efficacy in handling nonlinear data in high-dimensional space. However, current MKC methods have three primary issues: (1) Solely focus on clustering information while neglecting energy information and potential noise interference within the kernel; (2) The inherent manifold structure in the high-dimensional space is complex, and they lack the insufficient exploration of topological structure; (3) Most encounter cubic computational complexity, posing a formidable resource consumption challenge. To tackle the above issues, we propose a novel MKC method with shifted Laplacian on Grassmann manifold (sLGm). Firstly, sLGm constructs $r$-rank shifted Laplacian and subsequently reconstructs it, retaining the clustering-related and energy-related information while reducing the influence of noise. Additionally, sLGm introduces a Grassmann manifold for partition fusion, which can preserve topological information in the high-dimensional space. Notably, an optimal consensus partition can be concurrently learnt from above two procedures, thereby yielding the clustering assignments, and the computational complexity of the whole procedure drops to the quadratic. Conclusively, a comprehensive suite of experiments is executed to roundly prove the effectiveness of sLGm.

## CCS CONCEPTS

• **Computing methodologies → Machine learning**.

## KEYWORDS

Multiple kernel clustering, Shifted Laplacian, Grassmann manifold

## 1 INTRODUCTION

Clustering stands as a cornerstone in the realm of unsupervised learning, playing a vital role across various applications [25]. Its primary objective lies in partitioning similar data into the same cluster, thereby minimizing intra-cluster sample differences while maximizing inter-cluster differences [8]. In practical scenarios, the data landscape involves diverse sources, resulting in various clustering methods including graph learning based [23], kernel learning based [15] and subspace learning based [29] *etc.* Among them, the kernel learning based methods have obtained considerable attention owing to their efficacy in handing nonlinear data, which improves its separability by mapping nonlinear data into the reproducing

**Unpublished working draft. Not for distribution.**

Hilbert kernel space [13, 26]. Therein, multiple kernel-based clustering (MKC) can circumvent the choice of kernel function and the adjustment of kernel parameters, seamlessly integrating the information from each kernel. Typically, it involves constructing the kernel pool comprising base kernels, capturing the underlying structure from these kernels.

Broadly speaking, MKC methods can be roughly classified into two groups: spectral graph-based (SG) and kernel $k$-means-based (KKM). For the SG methods, the primary focus lies in learning a superb affinity graph within the kernel space, and thus use spectral clustering (SC) for obtaining the clustering assignments [4, 17]. For the KKM methods, they are customary to integrate base kernel into an optimal one, subsequently employing kernel $k$-means on this optimal kernel for acquiring the clustering assignments [10, 22]. Nevertheless, when confronted with a kernel matrix of $n \times n$, these two groups typically encounter cubic computational complexity, presenting a formidable challenge, especially in the context of large-scale datasets.

To deal with medium and large tasks, the MKC methods based on late fusion paradigm have been proposed. Such methods fuse the information of individual partitions to obtain the underlying shared kernel partition, significantly reducing the computational burden [19, 28]. For example, [21] employs orthogonal transformations to maximize the weighted base partition of individual kernels with a consensus partition. Building upon this foundation, [27] and [20] introduce local and global kernel maximization alignment respectively to delve into the structure embedded within the kernel. [7] proposes to combine the min-max scheme with the late fusion paradigm to simplify the objective function. Nevertheless, when obtaining the base kernel partition, these methods fail to simultaneously preserve energy and clustering information. In addition, in high-dimensional space, where the intrinsic manifold structure has the characteristics of bending and folding, they overlook the distance and topological information, resulting in suboptimal clustering performance.

To tackle the aforementioned problems, we propose a novel MKC method with shifted Laplacian on Grassmann manifold (sLGm). The illustration of sLGm is plotted in Fig. 1, we firstly construct the shifted Laplacian of individual kernel to simultaneously preserve energy-related and clustering-related information. Critically, to mitigate noise interference, the $r$-rank base kernel partition matrix of $r$-rank shifted Laplacian is constructed to derive the consensus underlying structure, and subsequently perform $r$-rank shifted Laplacian reconstruction to obtain the consensus partition. Considering the topological information preservation between individual $r$-rank partition and consensus partition, the Grassmann manifold is introduced to facilitate the acquisition of the optimal consensus partition. Finally, the consensus partition matrix is put into $k$-means for the assignments of clustering labels. Overall, the contributions of sLGm are as follows:

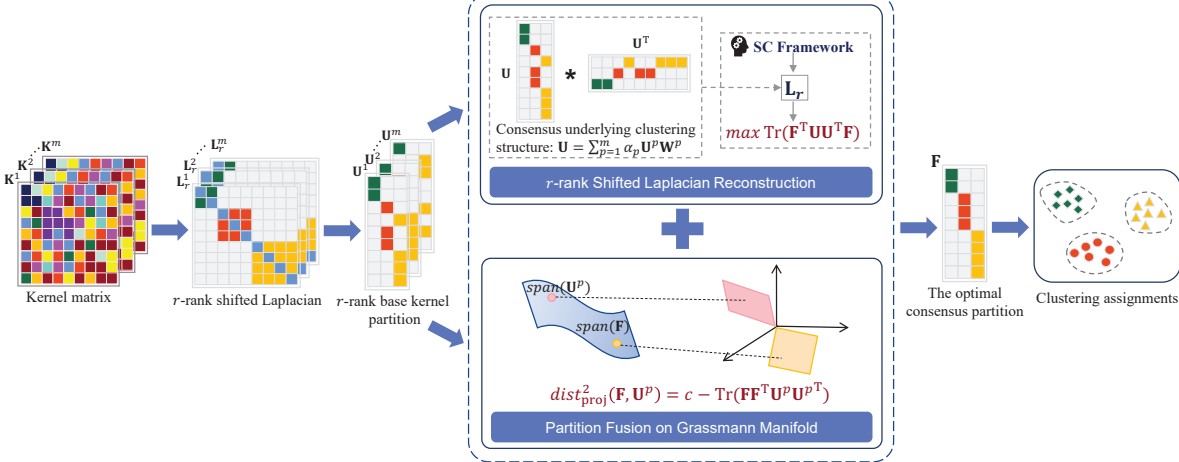

**Figure 1: Illustration of sLGm. Firstly, each $K^p$ can be treated as an affinity graph, and construct the $r$-rank shifted Laplacian $L_r^p$ and correspondingly yields the $r$-rank base kernel partition $U^p$, and then feed it into the two sub-frameworks. These two sub-frameworks are integrated to jointly learn an optimal F, thereby acquiring the clustering assignments.**

- sLGm fuses the individual $r$-rank base kernel partitions of the $r$-rank shifted Laplacian to perform reconstruction, which can minimize the noise interference while maximally preserving the clustering information and the reconstructed energy information.
- sLGm proposes to utilize the squared projection distance on the Grassmann manifold, to fuse the individual $r$-rank base kernel partitions and the consensus kernel partition, such that topological and heterogeneous information can be acquired simultaneously.
- sLGm integrates $r$-rank shifted Laplacian reconstruction and partition fusion on Grassmann manifold into a unified framework, where these two sub-frameworks jointly learn the optimal consensus partition for clustering, reducing the memory and computational complexity to $O(n)$ and $O(n^2)$ with the $n \times n$ kernel matrix.

## 2 RELATED WORK

### 2.1 Spectral Clustering with Laplacian Reconstruction

The concept of spectral clustering (SC) originates from graph theory, which can effectively capture the intrinsic structure of data via spectral decomposition or eigenvectors of Laplacian matrix. Specifically, for a data matrix $\mathbf{X} \in \mathbb{R}^{n \times d}$ with $n$ samples and $d$ features, its $k$-nearest affinity matrix is expressed by $\mathbf{Z} \in \mathbb{R}^{n \times n}$, whose edge between sample-pair can be defined by Gassuian kernel:

$$z_{ij} = \begin{cases} \exp\left(-\frac{\|\mathbf{x}_i - \mathbf{x}_j\|_2^2}{2\sigma^2}\right) & \text{if } \mathbf{x}_i \text{ and } \mathbf{x}_j \text{ are linked} \\ 0 & \text{otherwise} \end{cases} \quad (1)$$

where $\sigma$ is the width parameter, which controls how fast the similarity decays. And the diagonal elements of the degree matrix $\mathbf{D}$ of $\mathbf{Z} \in \mathbb{R}^{n \times n}$ are defined as $\mathbf{D}_{ii} = \sum_{j=1}^{n} \mathbf{Z}_{ij}$, thereby the Laplacian matrix $\mathbf{L} = \mathbf{D} - \mathbf{Z}$ can be obtained. Furthermore, the corresponding normalized Laplacian matrix $\mathbf{L_M}$ [16] can be defined as

$$\mathbf{L_M} = \mathbf{D}^{-\frac{1}{2}}(\mathbf{D} - \mathbf{Z})\mathbf{D}^{-\frac{1}{2}} = \mathbf{I}_n - \mathbf{D}^{-\frac{1}{2}}\mathbf{Z}\mathbf{D}^{-\frac{1}{2}} \quad (2)$$

Denoting $\mathbf{H} \in \mathbb{R}^{n \times c}$ as the indicator matrix with $c$ clusters, the objective function of the normalized SC [14] is expressed as

$$\min_{\mathbf{H}} \text{Tr}(\mathbf{H}^\top \mathbf{L_M} \mathbf{H}) \quad \text{s.t. } \mathbf{H}^\top \mathbf{H} = \mathbf{I}_c \quad (3)$$

The optimal $\mathbf{H}$ is constructed by taking the $c$ smallest eigenvectors of $\mathbf{L_M}$, and the clustering assignments are obtained by discretizing $\mathbf{H}$ with $k$-means. To capture complementary information from multiple views, [9, 24] perform Laplacian reconstruction to linearly integrate the base Laplacians, such that obtaining the optimal Laplacian. Such conceptual framework is formalized as follows:

$$\min_{\mathbf{H}^\top \mathbf{H} = \mathbf{I}_c, \boldsymbol{\beta}} \text{Tr}(\mathbf{H}^\top \mathbf{L_M^\beta} \mathbf{H})$$

$$\text{s.t. } \mathbf{L_M^\beta} = \sum_{i=1}^{v} \beta_d^i \mathbf{L_M^i}, \|\boldsymbol{\beta}\|_1 = 1, \boldsymbol{\beta} \geq 0 \quad (4)$$

where $\mathbf{L_M^i}$ is the $i$-th view Laplacian matrix, $\beta^i$ is the weight of the $i$-th Laplacian.

### 2.2 Late Fusion Clustering

In MKC tasks, direct execution of complex operations on the $n \times n$ kernel matrix will lead to substantial computational and storage overhead, making it impractical for real-world applications when $n$ is sufficiently large. Therefore, maintaining a light computational burden is an urgent demand for MKC tasks. Accordingly, Wang *et al.* [21] propose the late fusion clustering paradigm, which can reduce the computational and memory burden straightway. Specifically, they initially employ kernel $k$-means to obtain clustering partitions $\mathbf{H}^i$ for base kernels, and then use the rotation matrix $\mathbf{W}^i$ to linearly combine $\mathbf{H}^i$ to maximize alignment with the consensus cluster partition $\mathbf{H}^\star$. Concurrently, a regularization term is incorporated to ensure that consensus partition and the average partition $\mathbf{E}$ are

constrained within a comparable range. This late fusion clustering paradigm can be expressed as follows:

$$\max_{\mathbf{H}^{\star\top}\mathbf{H}^\star=\mathbf{I},\{\mathbf{W}^i\}_{i=1}^m,\boldsymbol{\eta}} \text{Tr}(\mathbf{H}^{\star\top}\mathbf{C}) + \zeta\text{Tr}(\mathbf{H}^{\star\top}\mathbf{E}),$$

$$\text{s.t. } \mathbf{W}^{i\top}\mathbf{W}^i = \mathbf{I}, \boldsymbol{\eta}^\top\boldsymbol{\eta} = 1, \eta_i \geq 0, \mathbf{C} = \sum_{i=1}^m \eta_i\mathbf{H}^i\mathbf{W}^i \quad (5)$$

where $\zeta$ is a trade-off parameter.

## 2.3 Shifted Laplacian

As specified in Eq. (3), the clustering information is encoded within the smallest eigenvectors of the Laplacian matrix $\mathbf{L_M}$. However, as illustrated in [3], when conducting Laplacian reconstruction such as Eq. (4), their best low-rank approximation is reconstructed by partially largest eigenpairs. In other words, the eigenvectors associated with the largest Laplacian eigenvalues are sufficient for achieving Laplacian reconstruction, indicating that the energy-related information is predominantly retained in these largest eigenvectors. This uncovers a crucial contradiction between reconstructing the Laplacian matrix and preserving clustering information. To solve this contradiction, [6] shift the normalized Laplacian, which is specifically expressed as follows:

$$\mathbf{L} = 2\mathbf{I} - \mathbf{L_M} = \mathbf{I} + \mathbf{D}^{-1/2}\mathbf{S}\mathbf{D}^{-1/2} \quad (6)$$

where $\mathbf{L}$ is denoted as shifted Laplacian (sL for short), which has several vital properties [6]: its smallest eigenvalues correspond to the largest eigenvalues of $\mathbf{L_M}$, and both of which fall in the interval $[0, 2]$; it is a symmetric positive semidefinite matrix like $\mathbf{L_M}$.

Leveraging these properties, it can be inferred that the optimal solution to Eq. (3) can be derived from $c$ largest eigenvectors of $\mathbf{L}$. Therefore, $\mathbf{L}$ can be effectively employed to facilitate Laplacian reconstruction, effectually encode clustering information and energy-related information.

## 3 PROPOSED METHOD

### 3.1 $r$-rank Shifted Laplacian Reconstruction

In MKC, considering the kernel $\{\mathbf{K}^1, \mathbf{K}^2, \ldots, \mathbf{K}^m\}$ with $c$ distinct clusters from $m$ kernels. Notably, each $\mathbf{K}^p \in \mathbb{R}^{n\times n}$ is treated as an affinity matrix, allowing for the exploration of abundant graph-based information embedded in the kernel matrix [18]. Peculiarly, we introduce $k$ as a regulator for the number of neighbors associated with each vertex, aiming at eliminating redundant edges. Therefore, the sL $\mathbf{L}^p$ of kernel $\mathbf{K}^p$ can be computed using Eq. (6).

As illustrated in Sec. 2.3, it is clear that not all eigenvalues make significant contributions to the process of Laplacian reconstruction. Simultaneously, the spectral graph theorem reveals that the key clustering information is predominantly contained within the $r$ rank largest eigenvalues of $\mathbf{L}^p$. In contrast, the remaining smallest eigenvalues carry a disproportionately higher amount of noise-related information rather than relevant clustering data, resulting

in a low signal-to-noise ratio (SNR). To tackle this issue, $\mathbf{L}^p$ is eigen-decomposed into $r$-rank part and cluster-irrelevant part:

$$\mathbf{L}^p = \mathbf{U}\boldsymbol{\Lambda}(\mathbf{U})^\top$$

$$= \begin{bmatrix} \mathbf{U}^r & \overline{\mathbf{U}^r} \end{bmatrix} \begin{bmatrix} \boldsymbol{\Lambda}^r & \mathbf{0} \\ \mathbf{0} & \overline{\boldsymbol{\Lambda}^r} \end{bmatrix} \begin{bmatrix} \mathbf{U}^r & \overline{\mathbf{U}^r} \end{bmatrix}^\top \quad (7)$$

$$= \mathbf{U}^r\boldsymbol{\Lambda}^r(\mathbf{U}^r)^\top + \overline{\mathbf{U}^r}\overline{\boldsymbol{\Lambda}^r} \cdot \overline{(\mathbf{U}^r)}^\top = \mathbf{L}_r^p + \overline{\mathbf{L}_r^p}$$

where $\boldsymbol{\Lambda}$ is the diagonal matrix formed by the eigenvalues of $\mathbf{L}^p$, i.e., $\boldsymbol{\Lambda} = \text{diag}(\lambda_1, \cdots, \lambda_n)$, and $0 \leq \lambda_n \leq \ldots \leq \lambda_1 \leq 2$, $\mathbf{U}$ is formed by the corresponding eigenvectors. $\boldsymbol{\Lambda}^r$ and $\mathbf{U}^r$ are matrices formed by $r$ largest eigenvalues and eigenvectors corresponding to $\boldsymbol{\Lambda}$ and $\mathbf{U}$, and $\overline{\boldsymbol{\Lambda}^r}$ and $\overline{\mathbf{U}^r}$ are matrices formed by their remaining $(n - r)$ elements. In particular, $r$ is set to $c \leq r \ll n$ to facilitate obtaining additional cluster-related information at the initial stage.

Consequently, for each $\mathbf{K}^p$, the corresponding $r$-rank shifted Laplacian $\mathbf{L}_r^p$ ($r$-sL for short) can be obtained. Mathematically, the corresponding $r$-rank base kernel partition $\mathbf{U}^p$ can be acquired by performing eigen-decomposition on $\mathbf{L}_r^p$, which can be explicitly expressed as

$$\max_{\mathbf{U}^p} \text{Tr}(\mathbf{U}^{p\top}\mathbf{L}_r^p\mathbf{U}^p) \quad \text{s.t. } \mathbf{U}^{p\top}\mathbf{U}^p = \mathbf{I}_c, \mathbf{U}^p \in \mathbb{R}^{n\times c} \quad (8)$$

The optimal solution $\mathbf{U}^p$ can be acquired by extracting the $c$ largest eigenvectors of $\mathbf{L}_r^p$. Inspired by [21], we introduce the rotation matrix $\mathbf{W}^p \in \mathbb{R}^{c\times c}$, which can align the base kernel partition $\mathbf{U}^p$ with the consensus underlying clustering structure $\mathbf{U}$. Usually, it can be described as follows:

$$\mathbf{U} = \sum_{p=1}^m \alpha_p\mathbf{U}^p\mathbf{W}^p \quad \text{s.t. } \alpha_p \geq 0, \boldsymbol{\alpha}^\top\boldsymbol{\alpha} = 1, \mathbf{W}^{p\top}\mathbf{W}^p = \mathbf{I}_c \quad (9)$$

Since Eq. (8) can be approximately transformed into $\min \|\mathbf{L}_r^p - \mathbf{U}^p\mathbf{U}^{p\top}\|_F^2$, it is further deduced that $\mathbf{L}_r^p = \mathbf{U}^p\mathbf{U}^{p\top}$. Therefore, the combined $\mathbf{L}_r$ can be reconstructed using $\mathbf{U}\mathbf{U}^\top$, and the sub-framework of $r$-rank shifted Laplacian reconstruction can be obtained via the SC framework regarding $\mathbf{L}_r$:

$$\max_{\mathbf{F}^\top\mathbf{F}=\mathbf{I}_c,(\mathbf{W}^p)^\top\mathbf{W}^p=\mathbf{I}_c} \text{Tr}(\mathbf{F}^\top\mathbf{U}\mathbf{U}^\top\mathbf{F})$$

$$\text{s.t. } \mathbf{U} = \sum_{p=1}^m \alpha_p\mathbf{U}^p\mathbf{W}^p, \alpha_p \geq 0, \boldsymbol{\alpha}^\top\boldsymbol{\alpha} = 1 \quad (10)$$

where $\mathbf{F}$ represents the consensus partition and its optimal solution can be derived through the eigen-decomposition of $\mathbf{L}_r$ (i.e., $\mathbf{U}\mathbf{U}^\top$). Noteworthily, the above process can be viewed as a late fusion paradigm, but the difference is that we use $r$-sL to form $\mathbf{U}^p$ and $\mathbf{U}$. As $r$-sL is capable of preserving both clustering and energy information while maintaining a high SNR, reconstructing it yields a notably strong performance.

### 3.2 Partition Fusion on Grassmann Manifold

The relationship between the base kernel partition $\mathbf{U}^p$ and the consensus kernel partition $\mathbf{F}$ can be established using Eq. (10). However, relying solely on Eq. (10) is insufficient, as it fails to account for the information of distance and heterogeneity in the high-dimensional space. Specifically, the intrinsic manifold structure in such space exhibits complexity with featuring folding and distortion characteristics, while Eq. (10) cannot adequately preserve the topological

structure and the spatial information across multiple kernels. To solve these issues, we introduce the squared projection distance on the Grassmann manifold, as specified in Definition 1 [2].

DEFINITION 1. *A Grassmann manifold $\mathcal{S}(x, n)$ is defined as a x-dimensional subspace in an n-dimensional space $\Re^n$, where each specific subspace can be mapped to a specific point on the manifold $\mathcal{S}$. Mathematically, each point on a manifold $\mathcal{S}(x, n)$ is broadened to be represented by an orthogonal matrix $\mathbf{G}$, whose columns can be extended to a x-dimensional subspace of the corresponding n-dimensional space, denoted $span(\mathbf{G})$. Suppose two subspaces as $span(\mathbf{G}_1)$, $span(\mathbf{G}_2)$, and their angle is defined as $\theta_i$, which can represent the geometric proximity of these subspaces, then the squared projection distance between $\mathbf{G}_1$ and $\mathbf{G}_2$ can be defined as*

$$\begin{aligned} \text{dist}_{\text{proj}}^2(\mathbf{G}_1, \mathbf{G}_2) &= \sum_{i=1}^{x} \sin^2 \theta_i = x - \sum_{i=1}^{x} \cos^2 \theta_i \\ &= x - \text{Tr}(\mathbf{G}_1 \mathbf{G}_1^\top \mathbf{G}_2 \mathbf{G}_2^\top) \end{aligned} \tag{11}$$

As stated in Definition 1, it is evident that the projection distance on Grassmann manifold can effectively reduce the subspace discrepancy. Accordingly, we bring $\mathbf{F}$ and $\mathbf{U}^p$ into Eq. (11) to minimize the kernel dissimilarity and facilitate information fusion:

$$\text{dist}_{\text{proj}}^2(\mathbf{F}, \mathbf{U}^p) = c - \text{Tr}(\mathbf{F} \mathbf{F}^\top \mathbf{U}^p \mathbf{U}^{p\top}) \tag{12}$$

To integrate kernel information from $\mathbf{U}^p$ and $\mathbf{F}$, while assigning an appropriate weight $\gamma_p$ to different $\mathbf{U}^p$ for acquiring the optimal $\mathbf{F}$, Eq. (12) can be rewritten as

$$\max_{\boldsymbol{\gamma}, \mathbf{F}} \sum_{p=1}^{m} \gamma_p \text{Tr}(\mathbf{F} \mathbf{F}^\top \mathbf{U}^p \mathbf{U}^{p\top})$$
$$\text{s.t. } \gamma_p \geq 0, \boldsymbol{\gamma}^\top \boldsymbol{\gamma} = 1, \mathbf{F}^\top \mathbf{F} = \mathbf{I}_c \tag{13}$$

In this way, Eq. (13) not only considers the heterogeneous information originating from multiple kernels, but also retains the spatial information and topological information, thereby accomplishing partition fusion. Consequently, by combining two sub-frameworks of $r$-rank shifted Laplacian reconstruction and partition fusion on Grassmann manifold, we derive the following objective function:

$$\max_{\boldsymbol{\gamma}, \boldsymbol{\alpha}, \mathbf{F}, \mathbf{W}^p} \text{Tr}(\mathbf{F}^\top \mathbf{U} \mathbf{U}^\top \mathbf{F}) + \lambda \sum_{p=1}^{m} \gamma_p \text{Tr}(\mathbf{F} \mathbf{F}^\top \mathbf{U}^p \mathbf{U}^{p\top})$$
$$\text{s.t. } \mathbf{U} = \sum_{p=1}^{m} \alpha_p \mathbf{U}^p \mathbf{W}^p, \gamma_p, \alpha_p \geq 0, \boldsymbol{\gamma}^\top \boldsymbol{\gamma} = 1, \tag{14}$$
$$\boldsymbol{\alpha}^\top \boldsymbol{\alpha} = 1, \mathbf{F}^\top \mathbf{F} = \mathbf{I}_c, (\mathbf{W}^p)^\top \mathbf{W}^p = \mathbf{I}_c$$

where $\lambda$ is a balance parameter. As depicted in Eq. (14), it primarily encompasses two sub-frameworks, these two complementary sub-frameworks mutually reinforce one another and collaboratively learn an optimal consensus partition $\mathbf{F}$, consequently yielding improved clustering performance.

## 4 OPTIMIZATION

### 4.1 The Optimal Solution

▷ *For $\boldsymbol{\gamma}$ and $\boldsymbol{\alpha}$:* When fixing other variables, Eq. (14) *w.r.t.* $\boldsymbol{\gamma}$ is expressed as

$$\max_{\boldsymbol{\gamma}} \sum_{p=1}^{m} \gamma_p \mathbf{X}_p \quad \text{s.t. } \gamma_p \geq 0, \boldsymbol{\gamma}^\top \boldsymbol{\gamma} = 1 \tag{15}$$

where $\mathbf{X}_p = \lambda \text{Tr}(\mathbf{F} \mathbf{F}^\top \mathbf{U}^p \mathbf{U}^{p\top})$, and the solution of $\boldsymbol{\gamma}$ is

$$\gamma_p = \frac{\mathbf{X}_p}{\sqrt{\sum_{p=1}^{m} \mathbf{X}_p^2}} \tag{16}$$

Likewise, when fixing other variables, the solution of $\boldsymbol{\alpha}$ is

$$\alpha_p = \frac{\mathbf{J}_p}{\sqrt{\sum_{p=1}^{m} \mathbf{J}_p^2}} \tag{17}$$

where $\mathbf{J}_p = \text{Tr}(\mathbf{F}^\top \mathbf{U}^p \mathbf{W}^p)$.

▷ *For $\mathbf{F}$:* Fix $\boldsymbol{\gamma}$, $\boldsymbol{\alpha}$ and $\mathbf{W}^p$, Eq. (14) *w.r.t.* $\mathbf{F}$ can be expressed as

$$\begin{aligned} \max_{\mathbf{F}^\top \mathbf{F} = \mathbf{I}_c} &\text{Tr}(\mathbf{F}^\top \sum_{p=1}^{m} \alpha_p \mathbf{U}^p \mathbf{W}^p (\sum_{p=1}^{m} \alpha_p \mathbf{U}^p \mathbf{W}^p)^\top \mathbf{F}) \\ &+ \lambda \sum_{p=1}^{m} \gamma_p \text{Tr}(\mathbf{F} \mathbf{F}^\top \mathbf{U}^p \mathbf{U}^{p\top}) \end{aligned} \tag{18}$$

which can be further rewritten as

$$\max_{\mathbf{F}^\top \mathbf{F} = \mathbf{I}_c} \text{Tr}(\mathbf{F}^\top \mathbf{H} \mathbf{F}) \tag{19}$$

where $\mathbf{H} = \sum_{p=1}^{m} \alpha_p \mathbf{U}^p \mathbf{W}^p (\sum_{p=1}^{m} \alpha_p \mathbf{U}^p \mathbf{W}^p)^\top + \lambda \sum_{p=1}^{m} \gamma_p \mathbf{U}^p \mathbf{U}^{p\top}$, and its optimal solution can be acquired via the largest $c$ eigenvectors of $\mathbf{H}$.

▷ *For $\mathbf{W}^p$:* When fixing other variables, the optimization *w.r.t.* $\mathbf{W}^p$ can be expressed via

$$\max_{(\mathbf{W}^p)^\top \mathbf{W}^p = \mathbf{I}_c} \text{Tr}(\mathbf{F}^\top (\sum_{p=1}^{m} \alpha_p \mathbf{U}^p \mathbf{W}^p)(\sum_{p=1}^{m} \alpha_p \mathbf{U}^p \mathbf{W}^p)^\top \mathbf{F}) \tag{20}$$

which can be simplified as

$$\max_{(\mathbf{W}^p)^\top \mathbf{W}^p = \mathbf{I}_c} \sum_{p=1}^{m} \text{Tr}((\mathbf{W}^p)^\top \mathbf{R}^p \mathbf{W}^p) \tag{21}$$

where $\mathbf{R}^p = \alpha_p^2 (\mathbf{U}^p)^\top \mathbf{F} \mathbf{F}^\top \mathbf{U}^p$, and the optimization *w.r.t.* $\mathbf{W}^p$ can be obtained via the largest $c$ eigenvectors of $\mathbf{R}^p$.

### 4.2 Complexity Analysis and Convergence Analysis

As laid out in Algorithm 1, the complexity of sLGm consists of four subproblems, *i.e.*, $\boldsymbol{\gamma}$, $\boldsymbol{\alpha}$, $\mathbf{F}$ and $\mathbf{W}_p$. For updating $\boldsymbol{\gamma}$ and $\boldsymbol{\alpha}$, their computational complexity are $O(4c^2mn)$; For updating $\mathbf{F}$ and $\mathbf{W}_p$, their computational complexity are $O(cn^2)$ and $O(c^3m)$, respectively. Since $c, m \ll n$, its computational complexity is approximated as $O(n^2)$. Accordingly, its memory complexity is $O(n)$.

In Algorithm 1, four subproblems need to be solved, all of which have optimal solutions, and it is evident that the objective of Algorithm 1 increases monotonically while optimizing one variable, and the remaining variables remain unchanged. In addition, according

**Algorithm 1** The algorithm of sLGm

**Input:**

Base kernel pools $\{\mathbf{K}^1, \mathbf{K}^2, \ldots, \mathbf{K}^m\}$; parameters $\lambda$; clusters number $c$; rank number $r$; neighbors number $k$;

**Output:**

The final clustering assignments;

**for** $p = 1$ to $m$ **do**

Construct $k$-nearest graph of $\mathbf{K}^p$;

Calculate its degree matrix $\mathbf{D}^p$ and the normalized sL $\mathbf{L}^p$ using Eq. (6);

Eigen-decompose $\mathbf{L}_p$ using Eq. (7) to acquire the $r$-sL $\mathbf{L}_r^p$;

Calculate the base kernel partition $\mathbf{U}^p$ using Eq. (8);

**end for**

**while** not converge **do**

Update $\boldsymbol{\gamma}$ and $\boldsymbol{\alpha}$ using Eq. (16) and Eq. (17), respectively;

Update consensus partition $\mathbf{F}$ using Eq. (19);

Update rotation matrix $\mathbf{W}^p$ using Eq. (21);

**end while**

Discretize $\mathbf{F}$ by $k$-means to acquire $c$ clusters.

to Lemma 1 [27], the upper bound of the proposed algorithm can be obtained.

LEMMA 1. *For $\forall i, j$,* $\mathrm{Tr}[(\beta_i \mathbf{A}_i \mathbf{B}_i)^\top (\beta_j \mathbf{A}_j \mathbf{B}_j)] \leq \mathrm{Tr}[(\mathbf{A}_i \mathbf{B}_i)^\top (\mathbf{A}_j \mathbf{B}_j)] \leq \frac{1}{2}(\mathrm{Tr}[(\mathbf{A}_i \mathbf{B}_i)^\top (\mathbf{A}_i \mathbf{B}_i)] + \mathrm{Tr}[(\mathbf{A}_j \mathbf{B}_j)^\top (\mathbf{A}_j \mathbf{B}_j)]),$ *if* $\mathbf{A}_i^\top \mathbf{A}_i = \mathbf{I}_c$ *and* $\mathbf{B}_i^\top \mathbf{B}_i = \mathbf{I}_c$, *we have* $\mathrm{Tr}[(\mathbf{A}_i \mathbf{B}_i)^\top (\mathbf{A}_i \mathbf{B}_i)] = \mathrm{Tr}(\mathbf{B}_i^\top \mathbf{A}_i^\top \mathbf{A}_i \mathbf{B}_i) = \mathrm{Tr}(\mathbf{I}_c) = c = \mathrm{Tr}[(\mathbf{A}_j \mathbf{B}_j)^\top (\mathbf{A}_j \mathbf{B}_j)].$

According to Lemma 1, the first term of Eq. (14) can be transformed into $\mathrm{Tr}(\mathbf{F}^\top \mathbf{U}\mathbf{U}^\top \mathbf{F}) \leq \frac{1}{2}(\mathrm{Tr}(\mathbf{F}\mathbf{F}^\top \mathbf{F}\mathbf{F}^\top) + \mathrm{Tr}(\mathbf{U}\mathbf{U}^\top \mathbf{U}\mathbf{U}^\top)) = \frac{1}{2}(\mathrm{Tr}(\mathbf{I}_c) + \mathrm{Tr}[\sum_{p,q,i,j=1}^{m} (\alpha_p \mathbf{U}_p \mathbf{W}_p)^\top (\alpha_q \mathbf{U}_q \mathbf{W}_q)(\alpha_i \mathbf{U}_i \mathbf{W}_i)^\top (\alpha_j \mathbf{U}_j \mathbf{W}_j)]) \leq \frac{c}{2}(m^4 + 1)$. In the same way, the second term is less than or equal to $\frac{\lambda c}{2}(m^2 + 1)$. As a whole, the proposed algorithm is less than or equal to $\frac{c}{2}((m^4 + 1) + \lambda(m^4 + 1))$. Therefore, Algorithm 1 has an upper bound and monotonically increasing, which can obtain a local maximum solution with convergence.

## 5 EXPERIMENT

### 5.1 Datasets

The experiments use eight datasets, encompassing various types. Specifically, they are BBCSports, Flower17, Handwritten, Caltech101, UCIdigits, Mfeat, YALE and SensVehicle. Preemptively, all kernel matrices associated with these datasets have been pre-computed and are available for download from a publicly accessible website. A brief description of all datasets is listed in Table. 1.

### 5.2 Experimental Settings

sLGm is compared with ten state-of-the-art methods, to summarize, they can be roughly divided into the following categories: SKC method, including B-SKKM; MVC methods, including LMVSC [5] and OMSC [1]; and MKC methods, including AMKKM, MR-MKKM [12], LFA [21], ONMSC-LF [9], SMKKM [11], LF-PGR [19] and LF-LKA [27]. For above all methods, their codes are downloaded from the provided website, and all parameters are adjusted according to their description. In particular, for all MVC methods, the kernel

**Table 1: Brief description of several datasets.**

| Dataset | Sample | Kenel | Cluster | Type |
|---|---|---|---|---|
| BBCSports | 544 | 2 | 5 | Text |
| Flower17 | 1360 | 3 | 17 | Image |
| Handwritten | 2000 | 6 | 10 | Graph |
| Caltech101 | 1530 | 25 | 101 | Image |
| UCIdigits | 2000 | 3 | 10 | Graph |
| Mfeat | 2000 | 12 | 10 | Graph |
| YALE | 165 | 5 | 15 | Image |
| SensVehicle | 1500 | 2 | 3 | Graph |

**Table 2: Complexity of the comparisons.**

| Method | Computational complexity |
|---|---|
| B-SKKM | $O(mn^2)$ |
| AMKKM | $O(mn^3)$ |
| MR-MKKM | $O(mn^2)$ |
| LFA | $O(nc^3 + mc^3)$ |
| LMVSC | $O(nm^3v + m^3v^3 + nc^2 + 2mvn + mnl)$ |
| OMSC | $O(n(l+c))$ |
| ONMSC-LF | $O(vn^3 + nc^2)$ |
| SMKKM | $O(n^3 + mn^2)$ |
| LF-PGR | $O(mcn^2 + cn^2)$ |
| LF-LKA | $O(nc^2 + mc^3)$ |
| sLGm | $O(4mnc^2 + c^3m + cn^2)$ |

matrix is treated as ordinary data for the algorithm input. In addition, to mitigate the impact of the randomness in $k$-means, we record the average results from 30 independent trials. To ensure a fair comparison, all methods are tested on the same device, and ACC, NMI and Purity are utilized as evaluation metrics.

In addition, for our method, there are three parameters to be adjusted, *i.e.*, $\lambda$, $r$ and $k$. For $\lambda$, set its value to [1e-5, 1e-4, ..., 100] to adjust dynamically. For $r$ and $k$, for convenience, two additional parameters $lrank$ and $kbur$ are introduced to adjust them, where $lrank = r \div c$ and $kbur = k \times c \div n$, with ranges of [1, 2, ..., 5] and [0.05, 0.1, ..., 1.5], respectively.

### 5.3 Experimental Results and Analysis

Table 3 presents the results of all methods across the eight datasets, from which the following observations can be acquired:

- Generally, sLGm consistently outperforms other methods on majority of datasets, notably surpassing the latest HF-MKKM, LF-PGR and LF-LKA. These findings suggest that sLGm is a viable method for dealing with nonlinear data.
- Compared with SKC, MKC methods are expected to yield superior clustering results. However, AMKKM, MR-MKKM and HF-MKKM methods exhibit suboptimal performance compared to B-SKKM on certain datasets. This indicates that the MKC methods need to choose a judicious and comprehensive learning strategy to exploit the information from base kernels. Notably, sLGm choose a preferable strategy to adeptly integrate information from base kernels, thereby attaining commendable clustering results.

**Table 3: Experimental results (%), where the optimal and suboptimal results are marked in red and blue, respectively.**

| Datasets | Metrics | B-SKKM | AMKKM | MR-MKKM | LFA | LMVSC | OMSC | ONMSC-LF | SMKKM | LF-PGR | LF-LKA | Ours |
|---|---|---|---|---|---|---|---|---|---|---|---|---|
| BBCSports | ACC | 76.65 | 66.18 | 66.18 | 77.45 | 66.84 | 74.93 | 83.82 | 67.40 | 80.51 | **86.58** | **96.32** |
| | NMI | 59.38 | 53.92 | 53.93 | 55.63 | 50.21 | 66.05 | **73.92** | 48.90 | 64.78 | 71.63 | **88.15** |
| | Purity | 79.59 | 77.20 | 77.21 | 76.27 | 85.47 | **91.73** | 84.01 | 73.07 | 81.25 | 86.58 | **96.32** |
| Flower17 | ACC | 42.05 | 51.02 | 58.82 | 61.16 | 62.28 | 63.88 | **65.74** | 59.38 | 62.42 | 63.97 | **67.75** |
| | NMI | 45.14 | 50.18 | 57.05 | 60.79 | 61.71 | 61.24 | **65.16** | 57.56 | 63.48 | 58.36 | **65.80** |
| | Purity | 44.63 | 51.98 | 60.51 | 62.32 | 62.72 | 63.30 | **66.99** | 60.55 | 62.42 | 64.19 | **69.54** |
| Handwritten | ACC | 86.41 | 78.03 | 93.07 | 94.15 | 93.76 | 94.05 | 96.85 | 93.26 | 92.25 | **97.65** | **97.70** |
| | NMI | 76.92 | 71.69 | 87.22 | 89.08 | 92.12 | 93.01 | 93.02 | 87.06 | 85.86 | **94.56** | **94.70** |
| | Purity | 86.55 | 77.50 | 93.10 | 94.05 | 93.20 | 94.05 | 96.85 | 93.26 | 92.25 | **97.65** | **97.70** |
| Caltech101 | ACC | 33.13 | 35.55 | 37.91 | 38.39 | 24.18 | 39.78 | **40.92** | 36.20 | 35.21 | 38.56 | **40.53** |
| | NMI | 59.06 | 59.90 | 61.47 | 61.65 | 52.65 | 51.24 | **63.96** | 60.68 | 60.02 | 62.42 | **63.60** |
| | Purity | 35.09 | 37.12 | 39.74 | 40.28 | 28.31 | 41.22 | **43.01** | 38.20 | 38.37 | 41.24 | **43.05** |
| UCIdigits | ACC | 75.40 | 88.75 | 90.40 | 88.60 | 75.45 | 92.80 | **97.15** | 90.47 | 82.90 | 95.50 | **97.55** |
| | NMI | 68.38 | 80.59 | 83.22 | 88.25 | 69.87 | 87.74 | **93.75** | 83.57 | 78.47 | 90.21 | **94.22** |
| | Purity | 76.10 | 88.75 | 90.40 | 88.90 | 78.25 | 93.40 | **97.15** | 90.47 | 82.90 | 95.50 | **97.55** |
| Mfeat | ACC | 86.00 | 95.20 | 92.55 | 95.15 | 96.70 | 95.85 | 97.00 | 94.05 | 93.75 | **97.85** | **98.25** |
| | NMI | 75.78 | 89.83 | 85.89 | **95.00** | 92.74 | 93.51 | 93.44 | 88.31 | 87.98 | 94.94 | **95.83** |
| | Purity | 86.00 | 95.20 | 92.55 | 95.05 | 96.70 | 95.85 | 97.00 | 94.05 | 93.75 | **97.85** | **98.25** |
| YALE | ACC | 47.12 | 38.97 | 60.00 | 62.42 | 53.94 | **64.88** | 63.46 | 55.67 | 62.42 | 63.03 | **66.09** |
| | NMI | 58.42 | 57.72 | 58.63 | 63.06 | 58.47 | 61.23 | **63.16** | 58.60 | 63.48 | 62.76 | **64.32** |
| | Purity | 57.58 | 53.94 | 60.00 | 62.46 | 63.64 | 64.30 | **64.85** | 56.00 | 62.42 | 63.64 | **66.73** |
| SensVehicle | ACC | 63.60 | 64.63 | 65.97 | **67.01** | 62.67 | 66.27 | **71.00** | 54.13 | 64.47 | 66.07 | **71.00** |
| | NMI | 18.57 | 21.15 | 21.75 | 22.64 | 20.20 | 22.13 | **28.49** | 11.28 | 20.64 | 23.28 | **30.63** |
| | Purity | 63.60 | 63.52 | 65.97 | 65.89 | 62.67 | 66.27 | **71.00** | 54.13 | 64.47 | **66.07** | **71.00** |

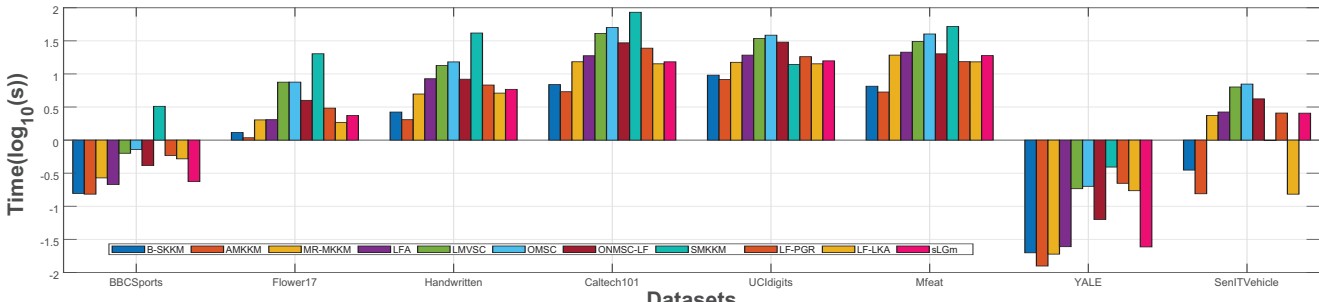

**Figure 2: Relative logarithmic time consumption comparison of eleven algorithms on eight datasets.**

- Compared with the MVC methods LMVSC and OMSC, sLGm achieves the best clustering performance, which is because they treat the kernel matrix as plaint data, can not fully explore the graph information, and have limitations for non-linear data processing.
- Compared with several MKC methods using late fusion, *i.e.*, LFA, ONMSC-LF, LF-PGR and LF-LKA, sLGm demonstrates superior performance. This can be attributed to two aspects, one is $r$-rank base kernel partition, enabling the preservation of both clustering and reconstruction information while mitigating the impact of noise. Another is the Grassmann manifold, which enhances the ability to capture topological

information. These two aspects contribute to the optimal consensus partition, which lead to an overall improvement in clustering performance.

## 5.4 Evaluation of Running Time and Parameters Sensitivity

**Running time:** The complexity of all comparisons are listed in Table 2, where $m, c, v, l, n$ represent the number of features, clusters, views, anchors and samples, respectively. Fig. 2 plots the comparison of time consumption on all algorithms. For convenience, the logarithm of time is taken as the ordinate, where the larger the value

is, the more time will be consumed. Based on this, it is evident that our algorithm exhibits relatively reduced time consumption compared to others, particularly in contrast to LF-PGR, which shares a comparable complexity. This discrepancy arises due to the fact that our algorithm encompasses less subproblems and their solution is simple. Furthermore, sLGm strategically performs sL with $r$-rank rather than full rank, resulting in a concomitant reduction in time consumption. This finding can provide preliminary evidence that sLGm is promising to handle large-scale data.

**Parameters sensitivity:** As previously stated, the algorithm involves three parameters: *i.e.*, $\lambda$, $lrank$ and $kbur$. To evaluate the sensitivity of these parameters, one variable is held constant over an extensive range while the others are systematically adjusted, as depicted in Fig. 3. These visualizations reveal that the clustering performance of all parameters is stable within a broad range, reflecting the insensitivity of sLGm to parameter variations.

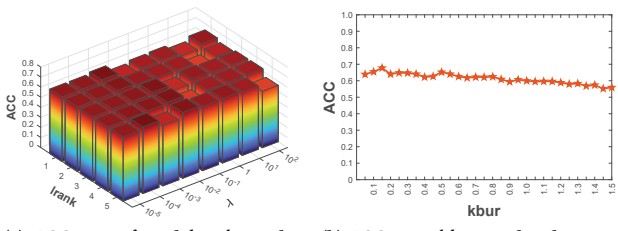

(a) ACC *w.r.t.* $\lambda$ and $lrank$ on the YALE dataset.

(b) ACC *w.r.t.* $kbur$ on the Flower17 dataset.

Figure 3: Parameters sensitivity.

## 5.5 Evaluation of the Effectiveness of $r$-sL

As the aforementioned experimental results presented, sLGm performs superior because the constructed sL can preserve cluster-related information, while the $r$-sL can remove noise-related information simultaneously. To verify the validity of sL and $r$-sL, we plot comparisons in terms of ACC on BBCSports and YALE, which is shown in Fig. 4. Note here that, to verify $r$-sL validity, the value of $r$ is adjusted, where $r = n$ means that $r$-sL is disabled. In addition, to visually show the clustering distribution results, the t-SNE visualization on the BBCSports dataset is plotted on Fig. 5. From Fig. 4 and Fig. 5, we can observe that: (1) ACC of using $r$-sL is significantly better than that of not using it; (2) ACC is relatively stable in a large range; (3) sLGm with $r$-sL has distinct clusters. The reason is that $r$-sL consists of largest eigenvalues that contain more cluster-related information, while small eigenvalues may contain more noise information hidden in the kernel matrix.

## 5.6 Evaluation of Ablation and Convergence

**Ablation evaluation:** To further verify the effectiveness of sLGm, ablation experiments are carried out on BBCSports, Flower17, YALE and SensVehicle by setting $\lambda = 0$ of Eq. (14). The corresponding results are listed in Table 4. The results distinctly indicate that when $\lambda = 0$, signifying the exclusion of partition fusion on Grassmann manifold, its clustering performance experiences an obvious reduction. This observation underscores the effectiveness of the

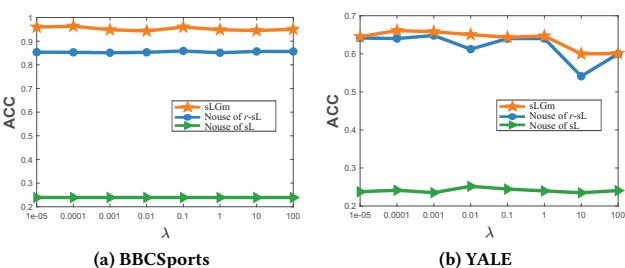

(a) BBCSports

(b) YALE

Figure 4: ACC and NMI in terms of sLGm, nouse of $r$-sL and nouse of sL for sLGm *w.r.t.* $\lambda$.

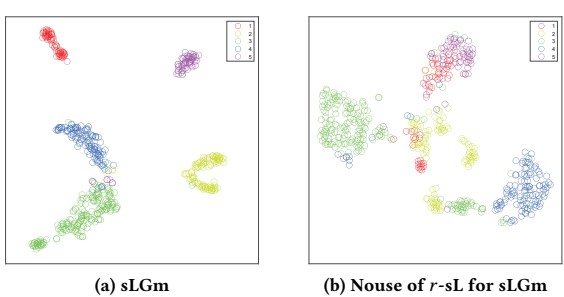

(a) sLGm

(b) Nouse of $r$-sL for sLGm

Figure 5: Visualization of clustering distribution with t-SNE on the BBCSports dataset.

utilization of the squared projection distance on the Grassmann manifold, as it effectively explores both topological and heterogeneous information in the high-dimensional complex space.

Table 4: Ablation experiments on part datasets.

| Datasets | | BBCSports | Flower17 | YALE | SensVehicle |
|---|---|---|---|---|---|
| ACC | sLGm | **96.32** | **67.75** | **66.09** | **71.00** |
| | sLGm$_{\lambda=0}$ | 80.70 | 65.48 | 64.58 | 70.53 |
| NMI | sLGm | **88.15** | **65..80** | **64.32** | **30.63** |
| | sLGm$_{\lambda=0}$ | 80.70 | 65.09 | 62.8 | 28.86 |
| Purity | sLGm | **96.32** | **69.54** | **66.73** | **71.00** |
| | sLGm$_{\lambda=0}$ | 84.19 | 68.13 | 65.18 | 70.53 |

**Convergence evaluation:** The convergence of the proposed algorithm has been analyzed in Sec. 4.2, and in order to verify the convergence experimentally, we draw a curve graph of the objective function value and clustering performance across iterations on UCIdigits and SensVehicle, as depicted in Fig. 6. It is observed that the objective function can converge to the stable value within five iterations. In addition, as the objective function converges, the corresponding clustering performance also stabilizes, indicating the effectiveness of the learnt consensus partition **F**.

## 6 CONCLUSION

In this paper, we propose a novel multiple kernel clustering method using shifted Laplacian on Grassmann manifold, *i.e.*, sLGm. Specifically, we take Laplacian as the breakthrough, construct $r$-rank

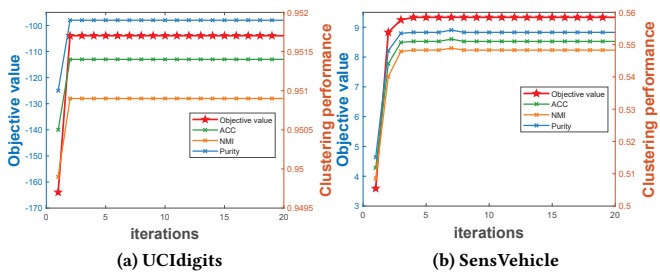

**Figure 6: Objective value and clustering performance *w.r.t.* iterations.**

shifted Laplacian and the corresponding *r*-rank base kernel partition to achieve reconstruction, which can retain the clustering information and energy information while remove the noise interference. In addition, the squared projection distance is used on the Grassmann manifold to further explore the topology structure. These two sub-frameworks jointly learn an optimal consensus partition to obtain the final clustering assignments. Compared with ten state-of-the-art clustering methods, and carried out a series of experiments on the benchmark datasets to comprehensively prove the effectiveness of sLGm.

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
