# OpenReview forum: "Multiple Kernel Clustering with Shifted Laplacian on Grassmann Manifold"
_acmmm.org/ACMMM/2024/Conference — MM2024 Poster_

### Official Review · Reviewer_VqKP · 2024-05-21

**Rating:** 4
**Confidence:** 3

**Summary:**

The paper proposes a novel Multiple Kernel Clustering (MKC) method called shifted Laplacian on Grassmann manifold (sLGm). This method aims to address three primary issues in existing MKC techniques: ignoring energy information and noise within the kernel, insufficient exploration of the high-dimensional manifold structure, and high computational complexity. The sLGm method constructs a rank-shifted Laplacian to retain clustering and energy-related information while reducing noise. It also introduces a Grassmann manifold to preserve topological information and facilitates partition fusion. The method promises reduced computational complexity and improved clustering performance. The authors validate their approach through extensive experiments on multiple datasets.

While the theoretical formulation and experimental results are solid, further detailed analysis and broader comparisons could provide a more comprehensive validation of the proposed method's advantages.

I'm inclined to accept it but may change my mind due to other reviewers' comments

**Strengths:**

The integration of shifted Laplacian and Grassmann manifolds for clustering is a novel approach. By addressing both clustering and energy information simultaneously while reducing noise, the paper introduces a significant innovation in the field of MKC.

The paper presents a clear mathematical derivation for the proposed method. The use of the Grassmann manifold for partition fusion is well-justified theoretically, adding robustness to the clustering process.

The experimental validation is thorough, involving multiple datasets. The results show that the proposed method outperforms several state-of-the-art techniques, demonstrating its effectiveness in various scenarios.

**Limitations:**

While the paper introduces a novel combination of techniques, the core ideas of using shifted Laplacian and Grassmann manifolds are not entirely new. Previous works have explored these concepts separately, eg. (Grassmann) manifold learning [1,2] shifted laplacian [3], and Late fusion[4]. The novelty primarily lies in their integration, which may be seen as incremental by some readers.

Although the experiments are extensive, the analysis could be more detailed. Specifically, it would be beneficial to include ablation studies to isolate the impact of each component (shifted Laplacian, Grassmann manifold) on the overall performance. This would provide clearer insights into the contributions of each part of the proposed method. The paper claims reduced computational complexity, but empirical evidence on large-scale datasets would better support this claim. Demonstrating scalability through experiments on very large datasets would enhance the argument for practical applicability.

[1] An attention-based framework for multi-view clustering on Grassmann manifold

[2] Multi-view Subspace Clustering on Topological Manifold

[3] Approximate shifted Laplacian reconstruction for multiple kernel clustering

[4] Late fusion multiple kernel clustering with proxy graph refinement

**Suitability:**

2

---

### Official Review · Reviewer_74Q9 · 2024-05-23

**Rating:** 5
**Confidence:** 4

**Summary:**

In this paper, the authors propose a novel multiple kernel clustering framework, which can retain the clustering information, energy information, and topology information while removing the noise interference. In general, it compared with ten comparative methods on eight datasets, the proposed method achieves good results.

**Strengths:**

Overall, the methodology and the framework of this paper is reasonable. Specifically, this paper has the following advantages:
(1) This paper has a corresponding derivation process for the proposed method, and lists the references of the phase management to prove the whole derivation process, which has good readability.
(2) This method proposed a 𝑟-rank shift Laplacian reconstruction technique, which can retain both cluster and energy information, which is not achieved by other methods. Also, Laplacian is used as the breakthrough to construct the shifted Laplacian operator, and manifold learning is introduced, these two can guide the learning of the optimal indicator matrix, thus improving the classification performance.
(3) In the EXPERIMENT section, the authors have carried out many experiments, and compared it with eight methods in related fields. Specifically, the authors show the effectiveness of sLGm from data analysis, chart analysis, figure analysis, and other aspects.

**Limitations:**

(1) The authors mention late fusion clustering in Sec 2.2, but do not describe much about late fusion when introducing the proposed method.
(2) Are the best parameters  ,   and   the same across different datasets? If not, some guidance on choosing these three parameters should be provided, which can provide instructions to the reader.
(3) Some sentences of this paper should be refined. For example, the sentence in right column of line 95 in page 1 is too long for the readers to understand.
(4) Some notations are not very clear. For example, in Sec 2.1, the notation $L_M^{\beta}$ should be interpreted concretely.
(5)Can you give a detailed explanation on how to reduce the computational complexity to $o(n^{2})$, whether the late fusion clustering  in Sec 2.2 is used, and how to reduce it? The necessary derivation and reasons need to be given.

**Suitability:**

3

---

### Official Review · Reviewer_AyY5 · 2024-05-24

**Rating:** 5
**Confidence:** 3

**Summary:**

The author proposes a novel multi-kernel clustering method based on the shifted Laplacian matrix on the Grassmann manifold. This model constructs 𝑟-rank Shifted Laplacian, preserving both clustering-related and energy-related information while reducing the impact of noise. Additionally, the model introduces a Grassmann manifold for partition fusion, which retains topological information in high-dimensional spaces. Extensive experimentation on real-world datasets validates the feasibility and effectiveness of the model.

**Strengths:**

1. The concept of the 𝑟-rank Shifted Laplacian is introduced innovatively, which not only preserves clustering-related and energy-related information but also reduces the impact of noise.
2. The experiments are quite extensive and sufficient, which effectively validate the effectiveness of the model.

**Limitations:**

1. The first and second terms of the model objective are very similar after expansion, could the differences be further explained theoretically?
2. The language quality of this paper still need improvement and enhancement. In Section 5.2, the second paragraph's description is somewhat colloquial and appears a bit disorganized.
3. How is the rotation matrix W initialized, and does its initialization affect the optimal significance of the model?

**Suitability:**

3

---

### Meta-Review · Area_Chair_SpXy · 2024-07-04

**Recommendation:** Accept (Poster)
**Confidence:** 5

**Metareview:**

The paper received three reviews, by reviewers who declare to be confident, in one case very confident. They all tend towards acceptance, although not enthusiastically  since their final (after rebuttal) recommendations are weak accept, weak accept, borderline accept.
The main limitations seem to be addressable in the camera ready, revised version of the paper, and after the rebuttal phase the reviewers comment that the paper appears to them acceptable.
I recommend accepting the paper for poster presentation.